

# Characteristics of ionospheric irregularities near the northern equatorial anomaly crest

Jinghua Li[1], Guanyi Ma[1,2], Klemens Hocke[3,4], Qingtao Wan[1], Jiangtao Fan[1], Xiaolan Wang[1]

[1] National Astronomical Observatories, Chinese Academy of Sciences, Beijing, China.

[2] University of Chinese Academy of Sciences, Beijing, China.

[3] Institute of Applied Physics, University of Bern, Bern, Switzerland

[4] Oeschger Centre for Climate Change Research, University of Bern, Bern, Switzerland

Corresponding to: G. Ma (guanyima@nao.cas.cn)

**Abstract**. This paper detects the ionospheric irregularities with rate of total electron content (TEC) change index, ROTI from GPS observation at Taoyuan (24.95 °N, 121.16 °E) for the solar medium and minimum years of 2003 and 2008 in the declining phase of cycle 23, the solar maximum of 2014 in solar cycle 24. Local occurrence rate (LOR) is proposed to clarify the characteristics of

5 the irregularities together with monthly occurrence rate (MOR) and ROTI maximum for 3 latitude belts, 20-23 °N, 23-26 °N, 26-29 °N, around the equatorial anomaly crest. MOR in May/June is larger than those in equinoxes in 2008 and 2003, which is different from that of equatorial plasma bubbles. In 2014 although MOR maximum is observed in equinoxes, the MOR in May and June is much larger than that in September. Moreover, MORs in May to August at higher latitude belt

26-29 °N are larger than those in lower latitude belts and smaller in the equinoxes. The latitudinal dependence of the LORs tends to be similar to that of MORs. Seasonal variations of LORs have a similar trend for different solar activities. Maximum LORs are observed in Feb/Mar and Sep/Oct, and moderate around June, which resemble those of plasma bubbles in seasonal variations, except for latitude belt 26-29 °N where maximum LORs are seen in May-Jul. The seasonal variation of

ROTI maximum conforms to that of the LOR. The results suggest that irregularities near the crest in May to August are mainly originated from nonequatorial process, which is more frequently happened but weaker than plasma bubble in both spatiotemporal scale and strength.

## 1 Introduction

The ionospheric irregularities are spatially irregular variation of electron density or
20 fluctuation of total electron content (TEC) with scale lengths from a few meters to several tens of kilometers. When radiowaves propagate through the irregularities, variations in signal strength happen due to refractive effect. This phenomenon is referred to as ionospheric scintillation, which



degrades both the performance of satellite communication and the precision of satellite navigation (Basu & Basu, 1981; Maruyama, 2002). Severe scintillation can lead satellite service interruption due to loss of lock in receivers. Moreover, TEC fluctuations in ionospheric irregularities are a major error to radio interferometers, differential Global Positioning System (GPS) and synthetic aperture radar (SAR) (Erickson et al., 2001; Zabotin and Wright, 2004; Afraimovich and Yasukevich,2008;Lee et al., 2011, Zheng et al., 2008).

Most intense ionospheric irregularities are those occurred in low-latitude regions near the magnetic equator. They were first recorded as the spreading of the traces of F layer echoes on ionograms with ionosonde. It has been since called equatorial spread F as well (Booker and Wells, 1938). Using the data from incoherent scattering radar, a plumelike irregular structure was found in a range-time-intensity diagram which provides the dynamic process of the irregularity evolution. It was then proposed that the irregularities are produced as low-density 'plasma bubbles' at the bottomside of the ionosphere and rose to an altitude of 1000 km (Woodman and La Hoz, 1976). The equatorial plasma bubbles (EPB) were directly confirmed by AE-C satellite in-situ measurements as regions of abrupt drop-out of electron density by two orders of magnitude with sizes of several tens of kilometers (McClure et al., 1977). Furthermore the optical imaging techniques probed two-dimensional structure of the plasma bubble (Weber et al., 1978; Mendilo and Baumgardner, 1982; Balan et al., 2018). It has been generally understood that plasma bubbles are generated at the bottom of the equatorial ionosphere by the generalized nonlinear Rayleigh-Taylor instability. While moving upward from the lower ionosphere into the higher density ionosphere, they extend along the magnetic flux tube to higher latitudes and often reach to the equatorial anomaly crest (Kelley and McClure, 1981; Ossakow, 1981).

The ionospheric irregularities have been studied with radar, satellite, airglow imager observations and simulations (Balan et al., 2018). Different observational techniques have their own advantage to reveal different aspects of the characteristics of the ionospheric irregularity. Since the civilian use of Global Navigation Satellite System (GNSS) the observation with ground-based dual-frequency GNSS receivers has become an important mean for ionospheric studies. While the fluctuation of phase and amplitude of GNSS signal, and hence scintillation, occur due to ionospheric irregularity, TEC also fluctuates. The rate of change of the TEC, termed ROT, and further, a rate of TEC index (ROTI) have been proposed as an indicator of presence of



ionospheric irregularity and a surrogate of TEC fluctuation (Aaron et al., 1996; Pi et al. 1997;

Basu et al., 1999). ROTI was used to investigate extensively the ionospheric equatorial bubbles or

irregularities at different time and over different regions for the last 20 years. Statistical and case

studies were successful in describing the occurrence features of the irregularity with local time,

seasonal variations, solar cycle and its geographical dependence. They confirm already existing

observational characteristics of the irregularity and give new insights into its global distribution

scenarios (Otsuka et al., 2006, Nishoka et al., 2008; Sripathi et al., 2018). Nevertheless, systematic

research of the ionospheric irregularity with ROTI is lacking at specific area for varying solar

activity. It is difficult to coordinate the available results since different research groups use

different definition in their studies (Mendillo et al., 2000; Otsuka et al., 2006; Nishioka et al.,

2008). Moreover, the strength of the irregularity or TEC fluctuation has not yet been addressed

quantitatively.

This paper aims to study statistical characteristics of plasma bubble associated irregularities

near the equatorial anomaly crest with continuous GPS observations for the solar minimum of

2008, medium of 2003 in the declining phase of cycle 23, and the solar maximum of 2014 in solar

cycle 24. Section 2 describes observation and data analysis method. Section 3 gives the results.

Section 4 deals with the discussion. Finally, conclusions are drawn in section 5.

## 70     2 Observation and data analysis

### 2.1 Observation

The GPS system composes of 24 to 32 satellites (on 6 planes) orbiting the Earth at an

inclination of 55 °and at a height of 20200 km. Each satellite broadcasts information on two

frequency carrier signals, which are 1.57542 GHz (referred to as $f_1$) and 1.2276 GHz (referred to

as $f_2$), respectively. Containing dozen or more parallel channels, a dual-frequency GPS receiver

is capable of receiving GPS signals from ~10 GPS satellites simultaneously. The observation is

from a dual-frequency GPS receiver with 30-s sampling rate taken by the International GNSS

Service (IGS) at Taoyuan (24.95 °N, 121.16 °E), Taiwan, which is the ionospheric equatorial

anomaly crest with frequent occurrence of irregularities. The data are from the solar medium of

2003, minimum of 2008 in the declining phase of cycle 23, and the solar maximum of 2014 in



solar cycle 24.

Figure 1 depicts a map of the IPP tracks of GPS satellites at 400 km observed by Taoyuan GPS receiver, with an elevation cutoff of 30 degrees, during 18:00~06:00 local time on 20th March 2003. The starting positions of the traces are marked with dots. The directions of the traces can be southward, northward, eastward, northeastward, or southeastward, except any westward orientation. The coverage of these traces is mainly within 20-29 °N in latitude and 116-126 °E in longitude. Due to the arrangement of the satellites and their periodic orbits, the coverage of the IPP traces is almost the same every day.

**2.2 ROTI calculation**

Under the irregularity "frozen-in" assumption, the characteristics of the ionospheric irregularities do not change within a short time when the ray path from a GPS satellite to a receiver traverses them. The TEC along the ray path would vary correspondingly due to the TEC fluctuation. By taking the difference between the slant TECs at two successive times, a rate of change of the TEC was defined as $ROT_i = (N_{Ti} - N_{Ti-1})/(t_i - t_{i-1})$ where $N_{Ti}$ is the slant TEC for the $i^{th}$ measurement at time $t_i$ (Aaron et al.,1996). Due to the random variation nature of the ionosphere, ROT is a time changing parameter. Further, ROTI, the standard deviation of ROT, is calculated to quantify the TEC fluctuation with time and space, $ROTI = \sqrt{\langle ROT_i^2 \rangle - \langle ROT_i \rangle^2}$ , where $\langle \rangle$ is the mean operator. With 30-s time series of GPS data, ROTI is generally calculated on a 5-min time window (Pi et al., 1997).

**2.3 Detection of irregularity traverse event**

The ionospheric irregularity is surveyed by a criterion as follows. The GPS observation with satellite elevation larger than 30 degrees is used in order to mitigate the effect of the multipath. ROTI is calculated on a 5-min time window with 11 successive data. An irregularity encounter is happened if ROTI is larger than a threshold. The threshold for the irregularity is the sum of the medium and 10 times of root mean square (rms) of all ROTIs between daytime 6:00 LT – 18:00 LT. If more than 20 consecutive ROTIs are larger than the threshold, the IPP is reckoned to

traverse the ionospheric irregularities, and an irregularity traverse event is identified. Another irregularity traverse event will be counted if it is encountered more than 1 h later than the first or preceding event. Although the threshold can be a little different day by day, it is in the range of 0.11-0.20 TECU/min for all the irregularity events in this study.

### 2.4 Occurrence rates definitions

Two kinds of occurrence rates are defined and studied. Monthly occurrence rate (MOR) is defined by the ratio of days with irregularity traverse observed to the days the observation is made. Higher monthly occurrence rate implies the irregularity happens frequently in daily scale. Local occurrence rate (LOR) is defined by the ratio of numbers of the irregularity encounters of all irregularity traverses to the numbers of all ROTI calculated between 17:00-7:00 LT. To compare with monthly occurrence rate, the local occurrence rate is counted based on one month. Higher local occurrence rate means the irregularity tends to exist with larger spatial and temporal scales.

### 3 Results and discussion

### 3.1 Start time of irregularity traverse

For the three years studied, 306 irregularity traverse events were identified. The distribution of the start time observed for the irregularity traverse is shown in Fig. 2. It is noteworthy that since the traces of the IPP can be any direction except westward orientation, the first time an irregularity encounter observed is not necessarily the onset time of the irregularity. Moreover, the irregularities observed in the same traverse event are not necessarily from the same source. From Fig. 2, most of the irregularity traverses are first observed after sunset and before midnight.

In 2003 there were 98 irregularity traverses, of which 71 (72%) were first seen at the time within 18:00-24:00 LT. In 2008, 33 out of 40 (82%) arose within 18:00-24:00 LT. It is 135 out of 168 events (80%) in 2014. The first observed times of irregularity traverses in 2014 have a prominent peak of 38 corresponding to 19:00-20:00 LT bin. The 38 irregularity traverses are mostly from Feb. and Mar. While there were only 7 (18%) irregularity traverses happened after midnight in 2008, there were 25 (26%) and 32 (19%) irregularity traverses observed later than midnight in 2003 and 2014, respectively.



### 3.2 Variation of Occurrence rates

Figure 3 shows the monthly occurrence rates (higher row) and the local occurrence rates (lower row) for solar minimum (2008), medium (2003) and maximum (2014). By dividing the area into 3 latitude belts, 20-23 ˚N, 23-26 ˚N, 26-29 ˚N, latitudinal dependence of the irregularity occurrence can be also studied. The seasonal variation of MOR has a clear dependence on solar activity. In 2008, the MOR has peaks of ~17%-25% and increases with latitude belt in May. There

is no irregularity observed in March and November for all the area. The MOR peaks in May/Jun and is generally larger in May to July than other months in 2003. The value of the peaks ranges from~35% to ~45%. Moderate maximum MORs can be seen in Feb/Oct, except for that of latitude belt 26-29. In 2014, the peaks range from ~45% to 72% in March; from ~55% to ~63% in June; and from ~10% to ~35% in September. It is noteworthy that the MOR for latitude belt 26-29 ˚N

has the largest maximum in June and no maximum in autumn. Concerning latitude dependence, the MOR increases with latitude in May/Jun. It is generally the largest in May to August for latitude belt 26-29 ˚N. It deceases with latitude in equinoxes and winter months, and this is especially true in 2014.

        It is obvious that the LOR behaves very different from the MOR. LOR is much smaller than

MOR. While the largest MOR can be 72% for latitude belt 20-23 ˚N in March 2014, the largest LOR at the same time and place is only ~15%. The LOR peaks dominantly in Feb/Mar and Sep/Oct except for the latitude belt 26-29 ˚N where the LOR shows larger maximum in June of both 2003 and 2014, and July of 2008. The LOR t is generally larger in Feb/Mar than Sep/Oct, although the result is reversed for the latitude belt 20-23 ˚N in 2003. In 2008, the LOR peaks in

February which is one month earlier than spring equinox and in October (the same in 2003) which is one month later than autumn equinox. Though the LORs tend to be larger in higher solar activity year, the differences of maximum LORs are small for different solar activity years. This suggests whether the irregularity at Taoyuan area happens frequently or not, their spatial and temporal scales are rather independent of solar activity. Similar to MOR's latitude dependence, the

LOR also decreases with latitude, except in May to August. The LOR is the largest in May to July for latitude belt 26-29 ˚N.



### 3.3 Variation of ROTI Maximum

ROTI generally starts from a low value, increases and reaches to a maximum, and then deceases with the evolution of irregularities. ROTI maximum for an irregularity traverse event can

be an indicator of irregularity strength. Figure 4 displays the variation of ROTI maximum for different latitude belts in the three different solar activity years. In solar minimum year of 2008, the ROTI maximum is very low and peaks in February with values of ~0.65-1.58 TECU/min for different latitude belts. It is around or below 0.50 TECU/min for other months. The differences of ROTI maximums among different latitude belts are generally small. In solar medium year of 2003,

ROTI maximum generally peaks at Feb/Mar and October with values of ~3.92-7.62 TECU/min and ~1.84-3.42 TECU/min, respectively. ROTI maximum for latitude belt 20-23 °N also has peaks at May and August. The ROTI maximum increases with latitude in March, it can also decrease with latitude in April-July, October and November. In solar maximum year of 2014, ROTI maximum has peaks of 4.61-8.29 TECU/min in Feb/Mar and of 2.25-4.86 TECU/min in Sep. The

ROTI maximum decreases with latitude in Feb/Mar. The largest ROTI maximum happens in latitude belt 23-26 °N in September. ROTI maximum increases with latitude in June and the peak value of 2.28 TECU/min for latitude belt 26-29 °N is comparatively large. The ROTI maximum for latitude belt 20-23 °N has a moderate peak of 4.55 TECU/min in November. There are also peaks in September, although the values of ROTI maximum are around 3.84 TECU/min, which is

smaller than that in May. ROTI maximum in 26-29 °N latitude belt is generally smaller than those in other latitude belts for the whole year except in June. Overall, the seasonal variation of ROTI maximum conforms to that of the LOR. ROTI maximum tends to be larger for higher solar activity.

### 3.4 ROTI maximum with solar activity

A scatter plot was attempted to examine in detail the relation of ROTI maximums for irregularity traverses with solar activity, as shown in Fig. 5. Here used is the radio flux at 10.7 cm (F10.7) as an indicator of the solar activity. The symbols of cross, circle and dot represent the year of 2003, 2008 and 2014, respectively. The colors in red, magenta, green and blue distinguish spring, summer, autumn and winter. In solar minimum of 2008, F10.7 is smaller than 80 sfu.



ROTI was generally around 0.50 TECU/min. The largest ROTI is 1.64 TECU/min in winter.
In solar medium and maximum years of 2003 and 2014, F10.7 is between 84-172 sfu and 86-208
sfu, respectively. ROTI is from 0.24 to 7.62 TECU/min in 2003. The largest value of ROTI is 8.82
TECU/min in winter of 2014, corresponding to 175 sfu. ROTI can be small (less than 1
TECU/min) however large F10.7 is, especially in summer. For F10.7 larger than 200 sfu in

summer, ROTI maximum is very small. ROTI can increase with F10.7 in all seasons of both 2003
and 2014. ROTI maximum tends to increase with F10.7 for those larger than 140 sfu, except for
summer in 2003 and 2014. As a whole the scatter was confined in a funnel. A large ROTI
maximum has a tendency to be related with a large F10.7, but the reverse is not always true.

## 4 Discussions

Low latitude irregularities are generally thought to be related with plasma bubbles generated
at the magnetic equator. The occurrence of plasma bubble in Asian region is known to be
maximum in equinoxes when sunsets in the conjugate E-regions are simultaneous (Tsunoda, 1985).
Nishioka et al. (2008) revealed higher monthly occurrence rate of plasma bubble around spring
equinox (summer solstice) than autumn equinox (winter solstice) from 2000 to 2006 with

ground-based GPS networks. Buhari et al. (2017) studied the occurrence rate of plasma bubble
with Malaysia Real-Time Kinematics Network (MyRTKnet) from 2008 to 2013. They found that
the occurrence day of EPB remains active during equinoctial months in low solar activity years.

In a morphology study of equatorial plasma bubbles during low and high solar activity years
in latitudes 13-17 °N over Indian sector, Kumar (2017) found maximum EPB occurrences during

the equinoctial months and minimum during the December solstice throughout 2007–2012 except
during the solar minimum years in 2007–2009. During 2007–2009, the maximum EPB
occurrences were observed in June solstice. Observations of equatorial and low latitude
irregularities with a meridional chain of ionosondes and GPS receivers in 2015 showed that
maximum EPB occurrences during the equinoctial months and minimum during the December

solstice throughout 2007–2012 except during the solar minimum years in 2007–2009. During
2007–2009, the maximum EPB occurrences were observed in June solstice (Kumar, 2017). As for
the equatorial anomaly crest, Lee et al. (2009) investigated the occurrence probabilities of



irregularity in solar maximum of 2000 with spread F, GPS phase fluctuations, and plasma bubbles
concurrently for the first time. The data used are mainly from the ionosonde and GPS receiver at
220 Chungli (24.9 °N, 121.2 °E) which is very close to Taoyuan, ROCSAT-1, and three GPS receivers
located at positions (14.0-14.6 °N, 121.0-121.1 °E) in Manila. To detect the irregularity traverse
event with GPS observation, they used an index Fp which is also based on rate of change of the
TEC, ROT, but takes the average median of ROT for all the satellites (refer to Mendillo et al.,
2000 for details). They found that the seasonal variations in GPS phase fluctuations at the crest
and the dip equator had similar trends in solar maximum of year 2000. They also showed the
range spread F (RSF) had similar occurrence probability to that of GPS phase fluctuations while
the frequency spread F (FSF) peaked in June as the spread F at mid latitude.

In Fig. 2, the MOR in solar maximum year of 2014 generally shows maximum values in
March, June, September, and November, respectively. The maximum value in March (June) is
230 much larger than that in September (November) for latitudes 20-26 °N. This is in agreement with
the results from GPS observations both at equatorial region and crest in solar maximum and high
activity years (Nishioka et al., 2008; Kumar, 2017; Lee et al., 2009). However, the MOR for
latitude 26-29 °N is larger in May-June than March in 2014, which is different from that of plasma
bubble. It is also noteworthy that the maximum values decrease with latitude, except that in May
to August which mainly increase with latitude. With ROTI from GPS observations in solar
maximum year of 2000 in Japan, Otsuka et al. (2006) showed similar results that the occurrence
rate peaks at equinoxes in 25 °N, but peaks at summer in 29 °N. In both solar medium 2003 and
minimum 2008, the MOR just shows maximum values in May and keeps larger values until
August. MOR increases with latitude, or MOR is the largest at 26-29 °N latitude belt implies that
the irregularity is encountered more frequently at higher latitude. Kumar (2017) also reported
maximum MOR in June observed with GPS receivers in latitudes 13-17 °N over India for
2007-2009. However, Buhari et al. (2017) found that the MOR of EPB over Malaysia (near
magnetic equator) was still active during equinoctial months in low solar activity years. So the
MOR of irregularity observed with Taoyuan GPS receivers is completely different from that of
245 plasma bubbles in solar medium and minimum year. This kind of seasonal variation of MOR is
similar to that of FSF at Chungli (24.9 °N) and that of spread F at midlatitudes in Japan (Lee et al.,
2009).





For the LOR, similar scenario of plasma bubble can be seen regardless of solar activity level. As shown in Fig. 2, the LOR in solar maximum year of 2014 generally decreases with latitude, except in June that the largest value is in latitude belt 26-29 ˚N. The equinoxes and solstices asymmetries are also there. In 2003 the LOR is more alike that of plasma bubble, although it has larger maximum value in October than March for latitude 20-23 ˚N. The two maximum and moderate peaks in 2008 are in February and November, respectively. They also decrease with latitude. However, the latitudinal feature of LOR in May to July is not the same as in other months. The largest LOR is always in 26-29 ˚N latitude belt for this period.

The physics behind the two occurrence rates behaviors can be explained with a schematic diagram of the earth's magnetic field lines at the meridian plane in Taoyuan as shown in Fig. 6. The intersections of the two solid bold magnetic field lines with the ionosphere at 400 km corresponds to latitudes of 20 ˚N and 29 ˚N, respectively. If the plasma bubble reaches specific latitude, the irregularity occurrence rate should be the same from the latitude to the magnetic equator. Due to the day to day variability, the plasma bubble occurrence rate should decrease with latitude. For both monthly and local occurrence rates decreasing with latitude, it implies that the irregularities are related with magnetic equator originated plasma bubbles. This can apply to most of the irregularities observed in equinoxes (Feb/Mar and Sep/Oct) and winter solstices months for all the three different solar activity years. However, the latitudinal variation of MOR in May to August cannot be attributed to plasma bubble. Since the MOR increase with latitude, it can be that the irregularities originated from nonequatorial process, for example, TID coming from mid latitude, as shown in Fig. 6. On the other hand, the latitudinal variation of LOR in May to August is not quite the same as that of MOR. LOR in the lowest latitude belt 20-23 ˚N is not always the smallest. LOR in latitude belt 26-29 ˚N is always the largest in May to July. It can be concluded that there can be overlaps of irregularities originated from both plasma bubble and nonequatorial processes. The nonequatorial processes are more frequently happened but weaker in spatiotemporal scale than plasma bubbles in May to August. The irregularities in 26-29 ˚N in May to July are mainly from nonequatorial processes.

Nonequatorial processes were also suggested for equatorial and low latitude irregularities with a meridional chain of ionosondes and GPS receivers in 2015, which showed that the postmidnight spread F during summer was weaker in strength and shorter in duration than





equatorial spread F mostly occurred in equinoxes and winter. Further the postmidnight spread F during summer is found to be stronger and earlier at low latitudes followed by their occurrence at the equator (Sripathi et al., 2018).

Figure 4 shows the variation of the irregularity's strength generally complies with its LOR in Taoyuan, although a one-to-one correspondence does not exist coming to those of latitude belt. In 2008, the ROTI maximum peaks and deceases with latitude in February. It is generally weak for the whole year, implying the contribution from plasma bubble is less than that from nonequatorial processes. In solar medium and maximum the irregularity associated with plasma bubble can be very strong not only in March, but also in Feb and April. The ROTI maximum in May to August in 2003 is the smallest in latitude belt 26-29 N, implying that the irregularity originated in nonequatorial processes is much weaker than plasma bubble. In 2014, more contribution from nonequatorial processes can be only seen in June. So, the nonequatorial origin irregularities are weaker (stronger) in strength than those related with plasma bubbles in solar medium and maximum (minimum) years. It is known that Perkins instability is responsible for the mid latitude irregularities (Perkins, 1973; Yokoyama et al., 2009). Otsuka et al. (2006) observed that the frequent occurrence of the irregularity in mid latitude in the summer night was usually accompanied by medium-scale traveling ionospheric disturbances (MSTID). Meridional observations from mid latitude to magnetic equator are needed to confirm and clarify the nonequatorial origin of the irregularities observed near equatorial anomaly crest.

The scatter plot of ROTI maximum versus F10.7 is rather dispersed, as shown in Fig. 5. However, confined in a funnel, a large ROTI maximum has a tendency to be related with a large F10.7, but the reverse is not always true. Higher solar activity is a necessary condition for the production of stronger plasma bubble related irregularities.

**5. Conclusion**

Making use of continuous GPS observations at Taoyuan (24.95 N, 121.16 E), characteristics of plasma irregularities near the ionospheric anomaly crest are studied for the solar minimum of 2008, medium of 2003 in the declining phase of cycle 23, and the solar maximum of 2014 in solar cycle 24. The irregularity is surveyed and detected with ROTI. Most of the irregularities are first





observed after sunset and before midnight. Local occurrence rate (LOR) is proposed to clarify the sources of the irregularities together with Monthly occurrence rate (MOR) and ROTI maximum for 3 latitude belts, 20-23 ˚N, 23-26 ˚N, 26-29 ˚N, within 116-126 ˚E longitude range. In 2014, while maximum MORs are observed in equinoxes, which is similar to that of plasma bubbles, the

MORs in May and June are much larger than those in September. In 2003 MORs are prominent in May-June, weaker maximums can be seen in February and October except for latitude belt 26-29 ˚N. In 2008 maximum MORs are only dominant in May. Moreover, MORs tend to be larger in higher latitude belts in May to August, while they tend to be smaller in higher latitude belts in other months. However, seasonal variations of LORs have a similar trend for different solar

activities. Maximum LORs are observed in Feb/Mar and Sep/Oct, and moderate around June, which resembles those of plasma bubbles in seasonal variations, except for latitude belt 26-29 ˚N where maximum LORs are seen in May-Jul. The latitudinal dependence of the LORs is not quite the same as that of MORs. The different behavior of MORs and LORs indicates that the irregularities in May to August are mainly originated from nonequatorial process, which is more

frequently happened but weaker than plasma bubble in both spatial and temporal scales. The seasonal variation of ROTI maximum conforms to that of the LOR. ROTI maximum tends to be larger for higher solar activity. Nonequatorial origin irregularities are weaker (stronger) in strength than those related with plasma bubbles in solar medium and maximum (minimum) years. Higher solar activity is a necessary condition for the production of stronger plasma bubble related

irregularities.

**Acknowledgement**

This research has been carried out under the support of the National Natural Science Foundation of China (NSFC No. 11873064) and National key Research Program of China "Collaborative Precision Positioning Project" (No.2016YFB0501900). IGS is acknowledged for free use of GPS

data. GM wishes to thank T. Maruyama for helpful discussion.



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




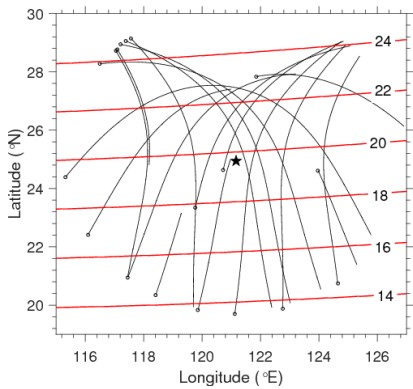

Fig. 1. IPP tracks of GPS satellites with 30 ° elevation cutoff from Taoyuan during 18:00~06:00 local time on 20th

March 2003. The star shows the location of the GPS receiver. The dots express starting positions of the tracks.

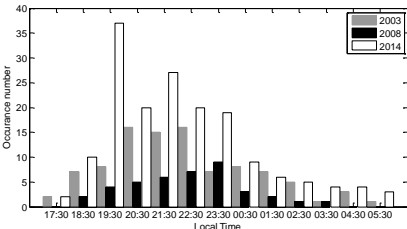

Fig. 2. Distribution of the first observed times for irregularity traverses. Most of the times fall in 19:00-24:00 LT

bin.

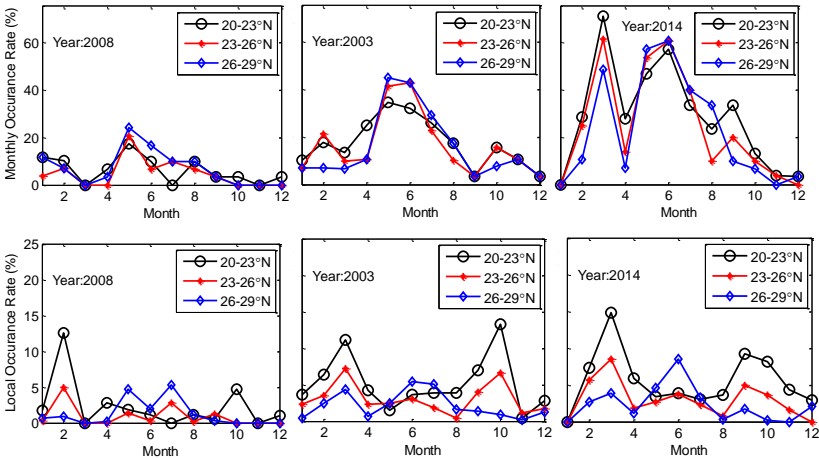

Fig. 3. Monthly and local occurrence rates of irregularities in equatorial anomaly crest.



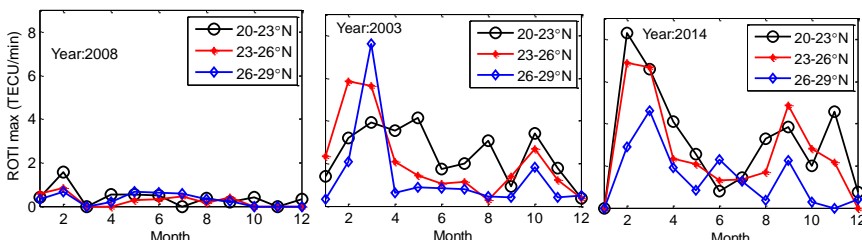

Fig. 4. Seasonal variations of ROTI maximum for irregularity traverses in equatorial anomaly crest.

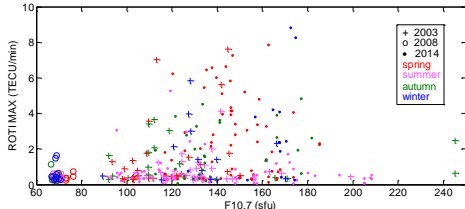

Fig. 5. Scatter plot of ROTI maximum for irregularity event versus F10.7 for different solar activity years.

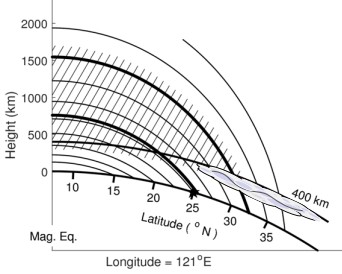

Fig. 6. A sketch of the plasma bubble and the earth's magnetic field lines at the meridian plane in Taoyuan.