# Peer review of "Characteristics of ionospheric irregularities near the northern equatorial anomaly crest"

_Annales Geophysicae, 2019_

## Referee Comment (RC1) · Anonymous Referee #1 · 22 May 2019

Review comments on paper "**Characteristics of ionospheric irregularities near the northern equatorial anomaly crest**" by Li et al., 2019

The paper attempts to discuss the occurrence of ionospheric irregularities using total electron content data derived from GPS observations over one location Taoyuan (24.95ºN,121.16ºE) during the years 2003, 2008 and 2014 based on the ROTI parameter. [On this note, the title should have specified the location of the study, otherwise in its present form, one may be led to believe that it is a global study near the northern equatorial anomaly crest].
Using one location's GPS data, the authors used TEC values at the ionospheric pierce points and categorised their analysis into three latitude bands of 3 degrees difference (20-23, 23-26 and 26-29).

Results are discussed in terms of local occurrence rate (LOR) and monthly occurrence rate (MOR) for both seasonal and daily analyses.
It is possible that the results may not be statistically significant when one station's data is used, the only difference being the separation of this data into three latitude regions. What stopped the authors from carrying out a detailed statistical analysis using data from various locations from mid-latitudes extending all the way to low/equatorial latitude region? In any case, during their discussions, they attributed some of their observations to non-equatorial processes possibly originating from mid-latitude regions.
The statement in the abstract that says "The results suggest that irregularities near the crest in May to August are mainly originated from nonequatorial process, which is more frequently happened but weaker than plasma bubble in both spatiotemporal scale and strength" is not supported by analyses/results and therefore is speculative.

Page 3, line 60; where they mention that systematic research of the ionospheric irregularity with ROTI in a specific... :  The authors should see papers by Mungufeni et al., (2016); Modeling of ionospheric irregularitiesduring geomagnetically disturbed conditions over African low-latitude region, Space Weather, 14, doi:10.1002/2016SW001446 and Mungufeni et al., (2016): Trends of ionospheric irregularities over African low latitude region during quiet geomagnetic conditions, JASTP,  261–267.

Pages 3-4: Details on how TEC (from where ROTI was derived) is calculated are missing. Please provide some statements about this and include the references where details of the algorithm/software used can be accessed.

Subsection 2.3:
Line 105, is the word "medium" supposed to be "median"? Under this subsection, the method of threshold detection is not clear and should be detailed. This should include a graphical demonstration to enable the reader understand the extent of data-length (in terms of time) which would typically fall within the time period chosen and what fraction fits the threshold definition.

On this, the text which mentions "ROTI is calculated on a 5-min time window with 11 successive data" is very difficult to understand. What is the meaning of 11 successive data?

On page 4, the authors considered ROTI values between 6:00-18:00 LT during irregularities' detection. However under subsection 2.4, the time has changed to 17:00-7:00 LT. Isn't this inconsistency?

Page 5, the statement "Moreover, the irregularities observed in the same traverse event are not necessarily from the same source". How do the authors come to this conclusion given that they are using data over one location?

Page 6, line 140, the authors say "There is no irregularity observed in March and November for all the area". This is a strong statement. Is this typically the case? How much data was available for the analysis during these months? Is there any literature available to support the authors' statement? I suggest that the authors perform similar analysis over a different location within the same region to confirm their statement.

Subsections 3.3 and 3.4: As I have mentioned in the previous comment, the division of the analysis into three latitude bands of 3 degrees separation based on data over one location could have its considerable limitations. Discussions in these subsections referring to maxima values of ROTI may therefore be very subjective. Based on this, the statistical results may not be statistically significant. It is suggested that the authors rather consider this location and perform the analysis without separation of different latitude regions, and have a look at a different location within the same region. Comparison of results and subsequent analysis based on two or more GPS locations is likely to provide reliable and realistic picture of irregularity occurrence. If the concern is about the satellites providing TEC data over a wider coverage area, the authors could limit their analysis to data with elevation threshold of 40-50 degrees.

Pages 8-9, lines 200-225: The authors are stating existing literature without tying it to their results/interpretation. This text therefore appears redundant in the paper.

Page 10, line 250 states "As shown in Fig. 2, the LOR in solar maximum year of 2014 generally decreases with latitude, ...". Firstly, there should be clarification whether LOR decreases with decreasing or increasing latitude. I notice that this clarification is required in the subsequent text as well. Secondly and perhaps most important is that the latitude range considered in this paper/analysis may be too small to make this conclusion.

How is Figure 6 generated?

Lines 260-270: The discussions here attributed irregularities to plasma bubbles and non-equatorial processes. However there is no evidence of each of these processes/mechanisms. The reader would expect authors to present occurrence of plasma bubbles and relate them to the irregularities discussed. There are a number of processes that take place in low latitudes including occurrence of plasma bubbles, scintillation, etc.

Lines 290-295, text talking about mid-latitude and suggestion that a study from mid-latitude to low magnetic equator is required.. I don't see why this wasn't done as GNSS receivers for this purpose are available.

There are a number of language usage errors that should be corrected.

In summary, what I see as a major shortcoming of this study is categorisation of data into three latitude blocks of 3 degrees separation and yet data is over one location. The subsequent statistical analysis is based on this which may be subjective. A similar analysis over two different locations may confirm the authors' results. A comprehensive study would be to look at generation of 2-D ROTI maps from where analysis can be performed. This would require consideration of a number of GNSS locations.

---

## Referee Comment (RC2) · Anonymous Referee #2 · 24 Sep 2019

The aim of the paper is to present characteristics of ionospheric irregularities at Taoyuan (Taiwan), a site under the northern crest of Equatorial Anomaly Ionization (EIA) using the ROTI from GPS observations during 2003, 2008, and 2014. They proposed the monthly occurrence rate (MOR) and the local occurrence rate (LOR) parameters to clarify the characteristics of the irregularities at three latitude belts. Even though the authors made an exhaustive work this paper should not be accepted in the present form due to the following main reasons: -first of all the authors couldn't explain clearly how the parameters MOR and LOR are able to point out the irregularity characteristics and to differentiate irregularities from equatorial origin from those with non-equatorial origin; - the authors didn't provide the position of the EIA crest in relation to the 3 latitude sectors for the 3 years. This EIA position depends of solar flux level;

[Figure]

- the time of occurrence of the non-equatorial irregularities is not provide; - the physical mechanisms, mainly for the non-equatorial irregularities are vaguely presented; - are the proposed parameters MOR and LOR created by the authors? This could be an original contribution from the paper, however at line 240 they mention that Kumar (2017) "also reported maximum MOR in June..". The authors should clarify this point. -at line 261-262 the authors stated: "Due to the day to day variability, the plasma bubble occurrence rate should decrease with latitude". Why? - MOR and LOR behaviors are presented repetitively at the "Results and discussion" section and at the "Discussion" section and this should be avoided to have a more objective paper; - the authors should discuss, at lines 295 to 300 as a suggestion, that even for high solar activity there are no irregularity events if the season is not favorable; - The English should be improved along the paper. Secondary but important improvements are suggested below: Line Corrections/suggestions 02 Inform dip latitude for Taoyuan 06 ..around the Equatorial Ionization Anomaly (EIA) 06-15 The text should be improved since MOR and LOR are not defined yet 15 near the EIA crest... 26 ..Differential Global Positioning System (DGPS) 28 Zheng et al., 2008 or 2009? 35 bubbles can easily reach even much more than 1000 km. Pls check this statement 44 equatorial ionization anomaly or use just EIA. 74 If the authors intend to describe GPS system, actually there are other frequencies 95 Aarons 104-106 Pls rewrite explaining better how the authors determine the threshold for the irregularity 105 average and 10 times... 107-109 Clarify the sentence Another irregularity....preceding event 117 Explain how: Higher local occurrence rate means the irregularity tends to exist with larger spatial and temporal scales. 120 Authors should use traverse irregularity (also along the paper) 127-128 Improve this phrase since it is not necessary to repeat 18:00-24:00 LT 131 The information that there are 38 traverse irregularities mostly from Feb. and Mar. cannot be seen from Figure 2. The authors should mention from which Figure they based to make this statement 132 Any reason to have less post-midnight irregularities during low solar activity? 137 Are the latitudinal bins in geographic coordinates? Please clarify 141 ...2003. In this year the value of... 157-159 Revise this statement since it is well

known that frequency and spatial and temporal Scales are solar flux dependent. Also MOR and LOR should clarify this statement and not to give origin to doubts: "suggests whether". Figure 4 shows low ROTI values for low solar flux 162 Variation of Maximum ROTI 164 Was a careful TEC data quality control done? If not false maximum ROTI could be generated. 172 ..in March and it decreases with. . . 175 Any reason for maximum ROTI decreasing with latitude in Feb/Mar in 2014 when it Increases during 2003? 184 ROTI maximum variation with solar flux 186-187 Here the radio flux at 10.7 cm (F10.7) was used as an . . . 203 Where are Nishioka et al (2008) data from? 219-221 Rewrite sentence since it is confusing 224 the EIA crest.. 228 Fig. 3 instead Fig. 2 231 and EIA crest 245 medium and minimum years. 249 Fig. 3 instead Fig. 2 250 in June when the largest. . . 253 February and November or February and October? 312 26-29 or 23-26.

---

## Referee Comment (RC3) · Anonymous Referee #3 · 30 Sep 2019

This manuscript presents statistical analysis of rate of TEC as observed from low latitude station (24.95o N) from Chinese sector. The statistics is obtained from a single GPS station using data from 3 different years of 2003, 2008 and 2014.

The manuscript has several lacunae with regard to analysis of data, result presentation and interpretation. Even considering this as a report, I could not find anything that adds to the existing knowledge on scintillation.

I provide my comments below, that boil down to rejection of the manuscript.

1. The geomagnetic latitude of the station is 18.20o north, which can not always be called the crest location under varying levels of solar activity. The crest of EIA has been used as a misnomer in several studies before, however, in reality this crest is

a dynamic latitudinal peak in TEC that varies even day-to-day, season-to-season and moves grossly towards dip equator during low solar activity periods. The peak in NmF2 may again differ from what one observes from TEC. Hence, for year 2008, the location cannot be granted for the crest of EIA. Authors shall mention this and carry necessary corrections in the manuscript.

2. 5-minute ROTI index has been calculated using estimated TEC. However, it has not been shown how TEC is estimated? If the GPS carrier phase data is used then how cycle slips are corrected which is an oft occurring event due to equatorial plasma bubbles passing over the site. Thus, ROTI itself can be ill-defined index to present the statistics. Result then become doubtful. Authors must clarify this issue by detailing.

3. Coming to the criterion used to declare traverse (occurrence)of EPB is not established by any means. Authors must provide 3-4 examples of estimation of TEC from RINEX data, then estimation of ROTI in panel below and then the criterion plotted along with the threshold. Thus, they may establish the validity for using it for all the data sets.

4. What are the physical rationales behind choosing 1-hr gape to reset the counter of EPB event? This seems gross qualitative measure. Now I cannot understand the statistics what it really represents?

5. MOR and LOR are ill-defined. There must be a plot to showcase how many days of observations were made in each month for all 3 years. Then MOR shall statistically significant and this must be quantified. At this level, nothing is known. In case of LOR, the number of irregularity counters are already proven wrong because of ill-defined criteria as mention in point 3 above. So how LOR is significantly true ?

6. I have studied several years of GPS observations using scintillation S4 index as well as ROTI index. The start time of irregularities can never be uniquely defined using a gross averaging index like ROTI? How much accurate will be this and this must be clarified?

7. Coming to the seasonal changes in variation of LOR and MOR, what is new that authors provide to a reader. All such variations are known. Amplitudes may vary that also is known. What is contribution of authors to add to existing knowledge is nowhere established.

8. How an average index of daytime solar radio F10.7 cm flux is related with ROTI amplitude?

9. Discussion section is highly flimsy. With help of some previous reports from very different durations than the present study covers, the discussion claims to the effect of solar activity of production of EPBs. This cannot be allowed in any sane scientific report. Production of EPBs depend upon two major physical processes that occur in post sunset duration over dip equator. One is triggering of EPB with seed perturbation and then non-linear growth of EPB. Then only it will be traversing over the low latitudes. Again, the fate of EPB depends upon background zonal drift, space weather events and electric field within the bubbles along with some secondary processes that produce a break the irregularity turbulence spectrum.

Hence just using a highly qualitative criteria based upon half-believable ROTI index cannot represent the truth that has occurred over the skies of this GPS station. Further, the latitudinal segregation of results is notional. Authors must use a greater number of sites to establishes any latitudinal behavior before commenting on fundamentals processes behind latitudinal changes in ROTI based criterion. Data is globally available and I am not finding any hindrance in using all the data to firmly establish what they wish to do.

Based on all above comments, I cannot suggest the manuscript to be even worth publication in discussion section of ANGEO. Editor may decide how much authors are willing to revise their paper and how much they will be able to really do with one station data?

ANGEOD

Interactive
comment

---

## Author Comment (AC1) · 28 Oct 2019

Thank you for your valuable and useful comments. According to your suggestions, we made major modification in this manuscript. Data from another GPS receiver located at (31.10°N, 121.20°E) was added. Unfortunately, no data can be obtained at lower latitudes than TWTF along the longitude of 121°E in the three years. The data from the two stations can provide the reliable and realistic results. In addition, we improve the English writing. In the following, we show the responses to the major comments one by one

1 The paper attempts to discuss the occurrence of ionospheric irregularities using total electron content data derived from GPS observations over one location Taoyuan

(24.95°N, 121.16°N) during the years 2003, 2008 and 2014 based on the ROTI parameter. [On this note, the title should have specified the location of the study, otherwise in its present form, one may be led to believe that it is a global study near the northern equatorial anomaly crest]. responseïijŽThe title has been changed to "Characteristics of ionospheric irregularities near the north EIA at 121°E" 2 Page 3, line 60; where they mention that systematic research of the ionospheric irregularity with ROTI in a specific... : The authors should see papers by Mungufeni et al., (2016); Modeling of ionospheric irregularitiesduring geomagnetically disturbed conditions over African low-latitude region, Space Weather, 14, doi:10.1002/2016SW001446 and Mungufeni et al., (2016): Trends of ionospheric irregularities over African low latitude region during quiet eomagnetic conditions, JASTP, 261–267. responseïijŽWe have read the papers and added the two references in the manuscript. 3 Pages 3-4: Details on how TEC (from where ROTI was derived) is calculated are missing. Please provide some statements about this and include the references where details of the algorithm/software used can be accessed. response: ROTI was derived from the relative slant TEC. The details on how to get it are stated in the sub section 2.2 of the manuscript. 4 Subsection 2.3: Line 105, is the word "medium" supposed to be "median"? Under this subsection, the method of threshold detection is not clear and should be detailed. This should include a graphical demonstration to enable the reader understand the extent of data-length (in terms of time) which would typically fall within the time period chosen and what fraction fits the threshold definition. response: The word "medium" has been corrected to "median". The method to get the threshold was added in subsection 2.3, equation (6). Figure A-1 shows an example of the traverse irregularity event detected by ROTI. 5 On this, the text which mentions "ROTI is calculated on a 5-min time window with 11 successive data" is very difficult to understand. What is the meaning of 11 successive data? response: This means the ROTI is calculated using 11 successive relative slant TEC. With the 30 seconds sampling interval, 11 successive data are in 5 minutes. We improved the description of ROTI in subsection 2.2. 6 On page 4, the authors considered ROTI values between 6:00-18:00 LT during irregularities' detection. However

under subsection 2.4, the time has changed to 17:00-7:00 LT. Isn't this inconsistency? response: The ROTI values between 6:00∼18:00 LT are used to calculate the threshold. The detection of the irregularities is based on the ROTIs during 17:00-7:00 LT and the threshold. We have improved the description. 7 Page 5, the statement "Moreover, the irregularities observed in the same traverse event are not necessarily from the same source". How do the authors come to this conclusion given that they are using data over one location? response: The statement is a speculation based on the large spatial range of IPPs. It may be inaccurate. We have deleted this in this manuscript. 8 Page 6, line 140, the authors say "There is no irregularity observed in March and November for all the area". This is a strong statement. Is this typically the case? How much data was available for the analysis during these months? Is there any literature available to support the authors' statement? I suggest that the authors perform similar analysis over a different location within the same region to confirm their statement. response: This is only description to Figure 3 in 2008. No data outage is in March of 2008 and the number of the observation days is 31.
* * *
**the 41st day in 2014 at TWTF**

ROTI (TECU/min)

UT

Figure A-1

**Fig. 1.**

---

## Author Comment (AC2) · 28 Oct 2019

Thank you for your valuable and useful suggestions. We made major modification by adding another GPS receiver located at (31.10°N, 121.20°E). The data from the two stations can prove the results in the last manuscript better. In addition, we improve the English writing. The response to your comment are attatched.

Please also note the supplement to this comment:
https://www.ann-geophys-discuss.net/angeo-2019-64/angeo-2019-64-AC2-supplement.pdf

**Supplement:**

Reply to Reviewer 2

Thank you for your valuable and useful suggestions. We made major modification by adding another GPS receiver located at (31.10 °N, 121.20 °E). The data from the two stations can prove the results in the last manuscript better. In addition, we improve the English writing.

Table 1 shows the responses to the major comments one by one. Table 2 presents the minor corrections.

Table 1 Response to the major comments

| No. | Comments | Modification/explanation |
|---|---|---|
| 1 | -first of all the authors couldnot explain clearly how the parameters MOR and LOR are able to point out the irregularity characteristics and to differentiate irregularities from equatorial origin from those with non-equatorial origin; | The definition of the MOR and LOR are given as equation (7) and (8) in this revised manuscript. The EPBs-induced irregularities can reach different latitudes from the dip equator in different events; therefore, the occurrence of these irregularities must decrease with latitudes in statistics. Otherwise, the irregularities are not from the EPBs, which are referred as non-equatorial process. By adding SHAO station (31.10 °N, 121.20 °E), obvious latitude dependence of MOR and LOR can be observed. The corresponding results and discussion are modified. |
| 2 | - the authors didn0t provide the position of the EIA crest in relation to the 3 latitude sectors for the 3 years. This EIA position depends of solar flux level. | I agree with you that the position of EIA crest depends on the solar flux level. It is closer to dip equator in 2008 than in 2003 and 2014. The aim of this paper is to present the characteristics of the ionospheric irregularities near the north EIA. As accurate description, we change the phase "in/near the north crest of EIA" to "near the north EIA". |
| 3 | -the time of occurrence of the non-equatorial irregularities is not provide; | SHAO station is located at (31.10 °N, 121.20 °E). The irregularities from this station were also studied from the occurrence time, occurrence rate, and the strength of TEC fluctuation. The irregularities at this station is not similar the EPBs', called non-equatorial irregularities. |

| No. | Comments | Modification/explanation |
|---|---|---|
| 4 | - the physical mechanisms, mainly for the non-equatorial irregularities are vaguely presented; | We focus on the characteristics of the irregularities in the low latitudes. By analyzing the latitude dependence of irregularities, the EPB and non-equatorial process are supposed as two contributions to the low latitude irregularities. The physical mechanisms is worthy to be studied, but not in the scope of this paper. |
| 5 | -are the proposed parameters MOR and LOR created by the authors? This could be an original contribution from the paper, however at line 240 they mention that Kumar (2017) "also reported maximum MOR in June..". The authors should clarify this point. | LOR is proposed in this paper. MOR has been used by many researchers. We revise the manuscript and describe this clearly. |
| 6 | --at line 261-262 the authors stated: "Due to the day to day variability, the plasma bubble occurrence rate should decrease with latitude". Why? | I am sorry for the unclear description. This sentence is modified as "The EPBs-induced irregularities can reach different latitudes from the dip equator in different events; therefore, the occurrence of these irregularities must decrease with latitudes in statistics." |
| 7 | MOR and LOR behaviors are presented repetitively at the "Results and discussion" section and at the "Discussion" section and this should be avoided to have a more objective paper; | By adding SHAO station's data, the results and discussion are improved to be reliable. |
| 8 | the authors should discuss, at lines 295 to 300 as a suggestion, that even for high solar activity there are no irregularity events if the season is not favorable; | The phase "necessary condition" was changed to "necessary but not sufficient condition" considering other mechanisms triggering the irregularities. |

Table 2 Response to the minor comments

| Comments | | Modification/explanation |
|---|---|---|
| line | Corrections/suggestions | |
| 02 | Inform dip latitude for Taoyuan | Dip latitude are added in the manuscript |
| 06 | ..around the equatorial Ionization Anomaly (EIA) | Accept the correction |
| 06-15 | The text should be improved since MOR and LOR are not defined yet 15 near the EIA crest. . . | The definition has been added in the abstract. |

| | Comments | Modification/explanation |
|---|---|---|
| 26 | ..Differential Global Positioning System (DGPS) | Accept the correction |
| 28 | Zheng et al., 2008 or 2009? | 2008 |
| 35 | bubbles can easily reach even much more than 1000 km. Pls check this statement | We change "1000 km" to "hundreds of the kilometers" according to the reference. |
| 44 | equatorial ionization anomaly or use just EIA. | Accept the correction |
| 74 | If the authors intend to describe GPS system, actually there are other frequencies | This paragraph has been removed because it is not necessary. |
| 95 | Aarons | Accept the correction |
| 104-106 | Pls rewrite explaining better how the authors determine the threshold for the irregularity | We give the definition of threshold. |
| 105 | average and 10 times. . . | Accept the correction. |
| 107-109 | Clarify the sentence Another irregularity. . ..preceding event | How to determine one irregularity traverse event is described in the revised manuscript. |
| 117 | Explain how: Higher local occurrence rate means the irregularity tends to exist with larger spatial and temporal scales. | The definition of LOR is presented in equation. And the relation between LOR and the spatiotemporal range is described. |
| 120 | Authors should use traverse irregularity (also along the paper) | Accept the correction |
| 127-128 | Improve this phrase since it is not necessary to repeat 18:00-24:00 LT | Section 3 is modified according to the results from the two stations. The description was improved. |
| 131 | The information that there are 38 traverse irregularities mostly from Feb. and Mar. cannot be seen from Figure 2. The authors should mention from which Figure they based to make this statement | The figure did not show the number. We try to describe the results quantitatively. But in the new manuscript, the results have been described according to the figures from the two stations. |
| 132 | Any reason to have less post-midnight irregularities during low solar activity? | In 2008, the number of the irregularity events is 40. And 18% events were after midnight, a little less than 19% in 2014 and 25% in 2003. It is a good question but now we cannot give reasonable explanation to the slight difference. |
| 137 | Are the latitudinal bins in geographic coordinates? Please clarify | In geographic coordinates. Manuscript has been modified according to the suggestion. |
| 141 | …2003. In this year the value of. . | Accept the correction. |

| Comments | | Modification/explanation |
|---|---|---|
| 157-159 | Revise this statement since it is well known that frequency and spatial and temporal Scales are solar flux dependent. Also MOR and LOR should clarify this statement and not to give origin to doubts: "suggests whether". Figure 4 shows low ROTI values for low solar flux | This statement is inaccurate. We revised this section according to the new figures based on the two stations. |
| 162 | Variation of Maximum ROTI | Accept the correction. |
| 164 | Was a careful TEC data quality control done? If not false maximum ROTI could be generated. | Yes, cycle slip and loss of lock are detected during the calculation of the relative slant TEC. |
| 172 | ..in March and it decreases with. . . | Accept the correction. |
| 175 | Any reason for maximum ROTI decreasing with latitude in Feb/Mar in 2014 when it Increases during 2003? | The dependence of ROTI maximum on latitudes (20~29N) is poor, and a good explanation has not been supposed. After adding another station to this paper, obvious difference of ROTI maximum can be found between the higher two latitude belts and the three lower ones. This is caused by the different strength of the irregularities in different latitudes. |
| 184 | ROTI maximum variation with solar flux | Accept the correction. |
| 186-187 | Here the radio flux at 10.7 cm (F10.7) was used as an . . . | Accept the correction. |
| 203 | Where are Nishioka et al (2008) data from? | The data are from the stations around the dip equator. This has been added to the manuscript. |
| 219-221 | Rewrite sentence since it is confusing | This sentence has been rewritten. |
| 224 | the EIA crest.. | Accept the correction. |
| 228 | Fig. 3 instead Fig. 2 | Accept the correction. |
| 231 | and EIA crest | Accept the correction. |
| 245 | medium and minimum years. | Accept the correction. |
| 249 | Fig. 3 instead Fig. 2 | Accept the correction. |
| 250 | in June when the largest. . . | Accept the correction. |
| 253 | February and November or February and October? | February and October. It has been corrected. |
| 312 | 26-29 or 23-26. | 26~29. |

---

## Author Comment (AC3) · 28 Oct 2019

Thank you for your attention and useful comments. The aim of the paper is to present characteristics of ionospheric irregularities near the EIA crest from GPS observations during 2003, 2008, and 2014. In this manuscript major modifications are as following: (1) Another GPS receiver located at (31.10°N, 121.20°E) was also used to study the irregularity. According to the latitudes of the IPPs, five latitudes belt are divided. The characteristics of the irregularity in the five latitude belts are studied and the latitude dependence is analyzed. (2) The figures from the two stations are plotted. The descriptions to the figures and the results from them are revised according to the new figures. (3) Discussion and conclusion are modified according to the results and the figures. (4) In addition, we improve the English writing. After the modification, the major contributions of this paper are summarized as: (1) Local occurrence rate (LOR) is proposed to describe the spatiotemporal range of the irregularities. (2) The monthly occurrence rate (MOR) is generally large in May/June than that in the equinox months. (3) LOR is the larger in the equinox months than in June for the lower latitudes. But for the higher latitudes, LOR is larger in June. (4) MOR and LOR in March and September/October decrease with the latitudes. But in June, they are large in the higher latitudes and small in the lower latitudes. (5) The characteristics of the irregularities in 20∼23°N and 23∼26°N are similar to the EPBs. But in the higher latitudes, they are different from the EPBs.

The responses to your comments are attatched.

Please also note the supplement to this comment:
https://www.ann-geophys-discuss.net/angeo-2019-64/angeo-2019-64-AC3-supplement.pdf

**Supplement:**

Reply to Reviewer 3

Thank you for your attention and useful comments.

The aim of the paper is to present characteristics of ionospheric irregularities near the EIA
crest from GPS observations during 2003, 2008, and 2014. In this manuscript major modifications
are as following: (1) Another GPS receiver located at (31.10 °N, 121.20 °E) was also used to study
the irregularity. According to the latitudes of the IPPs, five latitudes belt are divided. The
characteristics of the irregularity in the five latitude belts are studied and the latitude dependence
is analyzed. (2) The figures from the two stations are plotted. The descriptions to the figures and
the results from them are revised according to the new figures. (3) Discussion and conclusion are
modified according to the results and the figures. (4) In addition, we improve the English writing.

After the modification, the major contributions of this paper are summarized as: (1) Local
occurrence rate (LOR) is proposed to describe the spatiotemporal range of the irregularities. (2)
The monthly occurrence rate (MOR) is generally large in May/June than that in the equinox
months. (3) LOR is the larger in the equinox months than in June for the lower latitudes. But for
the higher latitudes, LOR is larger in June. (4) MOR and LOR in March and September/October
decrease with the latitudes. But in June, they are large in the higher latitudes and small in the
lower latitudes. (5) The characteristics of the irregularities in 20~23 °N and 23~26 °N are similar to
the EPBs. But in the higher latitudes, they are different from the EPBs.

The responses to the comments are presented in Table 1.

Table 1 Response to the comments

| No. | Comments | Modification/explanation |
|---|---|---|
| 1 | The geomagnetic latitude of the station is 18.20 °N north, which cannot always be called the crest location under varying levels of solar activity. The crest of EIA has been used as a misnomer in several studies before, however, in reality this crest is a dynamic latitudinal peak in TEC that varies even day-to-day, season-to-season and moves grossly towards dip equator during low solar activity periods. The peak in NmF2 may again differ from what one observes from TEC. Hence, for year 2008, the location cannot be granted for the crest of EIA. Authors shall mention this and carry necessary corrections in the manuscript. | The EIA did vary with the solar activity. TWTF is not always located in the EIA crest. It is more accurate to mention it as near the EIA. In the modified manuscript, we change "in the EIA crest" to "near the northern EIA". |

| No. | Comments | Modification/explanation |
|---|---|---|
| 2 | 5-minute ROTI index has been calculated using estimated TEC. However, it has not been shown how TEC is estimated? If the GPS carrier phase data is used then how cycle slips are corrected which is an oft occurring event due to equatorial plasma bubbles passing over the site. Thus, ROTI itself can be ill-defined index to present the statistics. Result then become doubtful. Authors must clarify this issue by detailing. | The method to obtain relative slant TEC is stated in the manuscript. During the calculation of ROTI, the difference between two adjacent slant TECs is used. The relative slant TEC and ROTI are calculated in every continuous arc. The cycle slip will cause ROTI outage in 5 minutes, but it does not affect the value of ROTI. The method to get ROTI referred the paper by Pi et al (1999). |
| 3 | Coming to the criterion used to declare traverse (occurrence) of EPB is not established by any means. Authors must provide 3-4 examples of estimation of TEC from RINEX data, then estimation of ROTI in panel below and then the criterion plotted along with the threshold. Thus, they may establish the validity for using it for all the data sets. | The criterions to calculate the threshold and detected the irregularity are described in the revised manuscript. An example is presented in the left panel of Figure A-1 to show the traverse irregularity event detected by ROTI. |
| 4 | What are the physical rationales behind choosing 1-hr gape to reset the counter of EPB event? This seems gross qualitative measure. Now I cannot understand the statistics what it really represents? | This is a good question. I agree with you. Sometimes the irregularity events are intermittent as shown in Figure A-1. 1 hour gape is based on a lot of examples. Whether other time gape is suitable is a question worth studying. In this manuscript we choose 1 hour to distinguish the irregularity after sunset or post midnight. |
| 5 | MOR and LOR are ill-defined. There must be a plot to showcase how many days of observations were made in each month for all 3 years. Then MOR shall statistically significant and this must be quantified. At this level, nothing is known. In case of LOR, the number of irregularity counters are already proven wrong because of ill-defined criteria as mention in point 3 above. So how LOR is significantly true ? | The definition of the MOR and LOR are presented by equations. The data outage is declared in the new manuscript. |

| No. | Comments | Modification/explanation |
|---|---|---|
| 6 | I have studied several years of GPS observations using scintillation S4 index as well as ROTI index. The start time of irregularities can never be uniquely defined using a gross averaging index like ROTI? How much accurate will be this and this must be clarified? | As you mentioned, ROTI has been used to study the irregularity popularly in these years. The accurate starting time is difficult to be determined by one way of observations for any event. Here we get the start time in statistics of hundreds of irregularity events. The coarse statistic is enough for analyzing the staring time in hour scale. |
| 7 | Coming to the seasonal changes in variation of LOR and MOR, what is new that authors provide to a reader. All such variations are known. Amplitudes may vary that also is known. What is contribution of authors to add to existing knowledge is nowhere established. | The main contribution of this paper is described in the first paragraph of the document. |
| 8 | How an average index of daytime solar radio F10.7 cm flux is related with ROTI amplitude? | The published paper showed that the occurrence of EPBs is related to the solar activity. Under magnetically quiet conditions, higher solar activity implies greater pre-reversal eastward electric field, earlier occurrence and earlier decay of EPBs (Fejer et al., 1999; Hysell et al., 2002). Solar flux number and the sunspot number have been as the input to the global ionospheric scintillation model (GISM) and the WBMOD ionospheric scintillation model. In this manuscript, we tried to analyze the relation between the F10.7 and ROTI maximum near the northern EIA. |

| No. | Comments | Modification/explanation |
|-----|----------|--------------------------|
| 9 | Discussion section is highly flimsy. With help of some previous reports from very different durations than the present study covers, the discussion claims to the effect of solar activity of production of EPBs. This cannot be allowed in any sane scientific report. Production of EPBs depends upon two major physical processes that occur in post sunset duration over dip equator. One is triggering of EPB with seed perturbation and then non-linear growth of EPB. Then only it will be traversing over the low latitudes. Again, the fate of EPB depends upon background zonal drift, space weather events and electric field within the bubbles along with some secondary processes that produce a break the irregularity turbulence spectrum. | The effect of solar activity on EPBs is described as stated above. |

[Figure]

                Figure A-1

---

## Author Comment (AC5) · 30 Oct 2019

Reply to reviewer 1

Thank you for your valuable and useful comments. According to your suggestions, we made major modification in this manuscript. Data from another GPS receiver located at (31.10 N, 121.20 E) was added. Unfortunately, no data can be obtained at lower latitudes than TWTF along the longitude of 121 E in the three years. The data from the two stations can provide the reliable and realistic results. In addition, we improve the English writing. In the following, we show the responses to the major comments one by one, and present the correction in Table 1.

Table 1 the response to reviewer 1

| No. | Comments | Modification/explanation |
|---|---|---|
| 1 | The paper attempts to discuss the occurrence of ionospheric irregularities using total electron content data derived from GPS observations over one location Taoyuan (24.95 N, 121.16 N) during the years 2003, 2008 and 2014 based on the ROTI parameter. [On this note, the title should have specified the location of the study, otherwise in its present form, one may be led to believe that it is a global study near the northern equatorial anomaly crest]. | The title has been changed to "Characteristics of ionospheric irregularities near the north EIA at 121 E" |
| 2 | Page 3, line 60; where they mention that systematic research of the ionospheric irregularity with ROTI in a specific... : The authors should see papers by Mungufeni et al., (2016); Modeling of ionospheric irregularitiesduring geomagnetically disturbed conditions over African low-latitude region, Space Weather, 14, doi:10.1002/2016SW001446 and Mungufeni et al., (2016): Trends of ionospheric irregularities over African low latitude region during quiet eomagnetic conditions, JASTP, 261–267. | We have read the papers and added the two references in the manuscript. |
| 3 | Pages 3-4: Details on how TEC (from where ROTI was derived) is calculated are missing. Please provide some statements about this and include the references where details of the algorithm/software used can be accessed. | ROTI was derived from the relative slant TEC. The details on how to get it are stated in the sub section 2.2 of the manuscript. |
| 4 | Subsection 2.3: Line 105, is the word "medium" supposed to be "median"? Under this subsection, the method of threshold detection is not clear and should be detailed. This should include a graphical demonstration to enable the reader understand the extent of data-length (in terms of time) which would typically fall within the time period chosen and what fraction fits the threshold definition. | The word "medium" has been corrected to "median". The method to get the threshold was added in subsection 2.3, equation (6). Figure A-1 shows an example of the traverse irregularity event detected by ROTI. |

| | | |
|---|---|---|
| 5 | On this, the text which mentions "ROTI is calculated on a 5-min time window with 11 successive data" is very difficult to understand. What is the meaning of 11 successive data? | This means the ROTI is calculated using 11 successive relative slant TEC. With the 30 seconds sampling interval, 11 successive data are in 5 minutes. We improved the description of ROTI in subsection 2.2. |
| 6 | On page 4, the authors considered ROTI values between 6:00-18:00 LT during irregularities' detection. However under subsection 2.4, the time has changed to 17:00-7:00 LT. Isn't this inconsistency? | The ROTI values between 6:00~18:00 LT are used to calculate the threshold. The detection of the irregularities is based on the ROTIs during 17:00-7:00 LT and the threshold.
We have improved the description. |
| 7 | Page 5, the statement "Moreover, the irregularities observed in the same traverse event are not necessarily from the same source". How do the authors come to this conclusion given that they are using data over one location? | The statement is a speculation based on the large spatial range of IPPs. It may be inaccurate. We have deleted this in this manuscript. |
| 8 | Page 6, line 140, the authors say "There is no irregularity observed in March and November for all the area". This is a strong statement. Is this typically the case? How much data was available for the analysis during these months? Is there any literature available to support the authors' statement? I suggest that the authors perform similar analysis over a different location within the same region to confirm their statement. | This is only description to Figure 3 in 2008. No data outage is in March of 2008 and the number of the observation days is 31. |

| 9 | Subsections 3.3 and 3.4: As I have mentioned in the previous comment, the division of the analysis into three latitude bands of 3 degrees separation based on data over one location could have its considerable limitations. Discussions in these subsections referring to maxima values of ROTI may therefore be very subjective. Based on this, the statistical results may not be statistically significant. It is suggested that the authors rather consider this location and perform the analysis without separation of different latitude regions, and have a look at a different location within the same region. Comparison of results and subsequent analysis based on two or more GPS locations is likely to provide reliable and realistic picture of irregularity occurrence. If the concern is about the satellites providing TEC data over a wider coverage area, the authors could limit their analysis to data with elevation threshold of 40-50 degrees. | Data from another GPS station named SHAO (31.10 N, 121.20 E) have been added to this paper to provide reliable and realistic picture of irregularity occurrence. |
|---|---|---|
| 10 | Pages 8-9, lines 200-225: The authors are stating existing literature without tying it to their results/interpretation. This text therefore appears redundant in the paper. | We rewrite section 4 and tie the literature to our results and interpretation. |
| 11 | Page 10, line 250 states "As shown in Fig. 2, the LOR in solar maximum year of 2014 generally decreases with latitude, ...". Firstly, there should be clarification whether LOR decreases with decreasing or increasing latitude. I notice that this clarification is required in the subsequent text as well. Secondly and perhaps most important is that the latitude range considered in this paper/analysis may be too small to make this conclusion. | Figure 2 is written by a mistake. It has been modified to Figure 3. The latitude dependence is more clear and reliable after SHAO station is used. The discussion has major modifications. |
| 12 | How is Figure 6 generated? | This figure is not very useful to the explanation after SHAO station is added. And it is removed from the manuscript. |

| 13 | Lines 260-270: The discussions here attributed irregularities to plasma bubbles and non-equatorial processes. However there is no evidence of each of these processes/mechanisms. The reader would expect authors to present occurrence of plasma bubbles and relate them to the irregularities discussed. There are a number of processes that take place in low latitudes including occurrence of plasma bubbles, scintillation, etc. | Here the word "plasma bubbles" means the equatorial plasma bubbles (EPBs). The EPBs-induced irregularities can reach different latitudes from the dip equator in different events; therefore, the occurrence of these irregularities must decrease with latitudes in statistics. Otherwise, the irregularities are not from the EPBs, which are referred as non-equatorial process. By adding SHAO station (31.10 °N, 121.20 °E), obvious latitude dependence of MOR and LOR can be observed. |
|----|----|----|
| 14 | Lines 290-295, text talking about mid-latitude and suggestion that a study from mid-latitude to low magnetic equator is required. I don't see why this wasn't done as GNSS receivers for this purpose are available. | SHAO station has been added according to your suggestion. |
| 15 | There are a number of language usage errors that should be corrected. | We tried to improve the English writing in the new manuscript. |

[Figure]

Figure A-1